Punicalagin, a pomegranate compound, induces apoptosis and autophagy in acute leukemia

Subkorn Paweena
Norkaew Chosita
Deesrisak Kamolchanok
Tanyong Dalina dalina.itc@mahidol.ac.th
Department of Clinical Microscopy, Faculty of Medical Technology, Mahidol University , Nakhon Pathom , Thailand
Silva Pedro
Electronic publication date: 2021 Nov 2
Publication date: 2021
Volume: 9
Electronic Location ID: e12303
Received 2021 Mar 22; Accepted 2021 Sep 22
Copyright: ©2021 Subkorn et al.
Copyright year: 2021
Copyright holder: Subkorn et al.
License: This is an open access article distributed under the terms of the Creative Commons Attribution License, which permits unrestricted use, distribution, reproduction and adaptation in any medium and for any purpose provided that it is properly attributed. For attribution, the original author(s), title, publication source (PeerJ) and either DOI or URL of the article must be cited.
License URL: https://creativecommons.org/licenses/by/4.0/

Keywords: Punicalagin, Apoptosis, Autophagy, Leukemia, Pomegranate

Funding: The Royal Golden Jubilee Ph.D. scholarship from the Thailand Research Fund PHD/0072/2561 The present study was supported by the Royal Golden Jubilee Ph.D. scholarship (grant no. PHD/0072/2561) from the Thailand Research Fund. The funders had no role in study design, data collection and analysis, decision to publish, or preparation of the manuscript.

==============================
Background

Punicalagin is the major phenolic compound found in pomegranate peels. It has several reported medical benefits, including antioxidant, anti-inflammatory, and anticancer properties. The present study investigated the anti-leukemic effects and the molecular mechanism of punicalagin on NB4 and MOLT-4 leukemic cell lines.

Methods

Leukemic cells were treated with punicalagin and cell viability was determined using MTS assay. Apoptosis and autophagy were analyzed by flow cytometry using Annexin V-FITC/PI and anti-LC3/FITC antibodies staining, respectively. Apoptotic and autophagic mRNA expression were determined using reverse transcription-quantitative PCR. STITCH bioinformatics tools were used to predict the interaction between punicalagin and its proposed target proteins.

Results

Results indicated that punicalagin decreased NB4 and MOLT-4 cell viability in a dose-dependent manner. Punicalagin, in combination with daunorubicin, exhibited synergistic cytotoxic effects. Punicalagin induced apoptosis through the upregulation of caspase-3/-8/-9, Bax and the downregulation of Bcl-2 expression. Punicalagin also promoted autophagy via the downregulation of mTOR and the upregulation of ULK1 expression. Cyclooxygenase-2 and toll-like receptor 4 were found to be involved in punicalagin-induced cell death in punicalagin-targeted protein interactions.

Conclusions

These results suggest that punicalagin exerts cytotoxic activities by suppressing proliferation and promoting apoptosis and autophagy by activating the caspase cascade, altering Bax and Bcl-2, and regulating autophagy via mTOR/ULK1 signaling.

Introduction

Leukemia is characterized by an increased production of hematopoietic stem cells in the bone marrow to impairs hematopoiesis (Davis, Viera & Mea, 2014). There were 60,530 new leukemia cases and 23,100 deaths in 2020 in the United States (Siegel, Miller & Jemal, 2020). Treatments for leukemia include chemotherapy, stem cell transplantation, and targeted treatment of the disease (Greaves, 2016). Chemotherapy, which is meant to reduce the number of leukemic cells, has various side effects including nausea, hair loss, and changes in appetite (Ramirez et al., 2009). The relapse of the disease may be associated with conventional chemotherapy (Yilmaz et al., 2019). Therefore, there has been a strong focus on developing alternative medicine using natural compounds and herbs for improving therapeutic efficacy (Hwang et al., 2019; Saedi et al., 2014).

Punicalagin (Fig. 1) is the major active compound present in pomegranate peel (Khwairakpam et al., 2018). It exhibits antioxidant (Abid et al., 2017), antibacterial (Xu et al., 2017), antiviral (Lin et al., 2013) and anti-inflammatory activities (BenSaad et al., 2017). There is some concern over the bioavailability and concentration of punicalagin and some in vivo bioavailability studies have been conducted (Espín et al., 2007). Ellagic acid is a product of hydrolysis from punicalagin that is metabolized by intestinal microflora with the remaining punicalagin to produce urolithins. The metabolites are then absorbed in intestinal cells and circulated in the blood stream (Espín et al., 2007; Vora, Londhe & Pandita, 2015). Oral consumption of punicalagin by rats has shown that punicalagin is present in plasma at concentrations of approximately 30 µg/ml (Cerdá et al., 2003).

Figure 1 Chemical structure of punicalagin.

Punicalagin has been reported to have anticancer properties against various cancer cell lines including colon (Larrosa, Tomás-Barberán & Espín, 2006), ovarian (Tang et al., 2016), prostate (Adaramoye et al., 2017) and lung (Berköz & Krośniak, 2020) cancer cells due to the inhibition of proliferation and the induction of cell cycle arrest and apoptosis. In vivo, punicalagin demonstrated anti-angiogenic effects by inhibiting blood vessel growth in the chorioallantoic membrane of chickens (Adaramoye et al., 2017). Punicalagin has been shown to possess anti-leukemic activities by suppressing proliferation and promoting apoptosis of HL-60 leukemic cells through caspase-3 activation (Chen et al., 2009). Pomegranate juice and peel extracts contain punicalagin, which is responsible for the induction of apoptosis, S phase cell cycle arrest, and inhibiting proliferation in CCRF-CEM, MOLT-3, HL-60 and THP-1 cells (Dahlawi et al., 2013; Tamborlin et al., 2020).

The induction of cell death through apoptosis, autophagy, and necrosis are key in cancer therapy (Mishra et al., 2018). Punicalagin has been shown to activate the intrinsic apoptosis pathway via the down-regulation of Bcl-XL and the activation of caspase-9 and caspase-3 in Caco-2 colon cells. The activation of the extrinsic pathway via caspase-8 and caspase-3 activation in PC-3 prostate cells can also induce apoptosis (Adaramoye et al., 2017; Larrosa, Tomás-Barberán & Espín, 2006). Punicalagin was also able to induce apoptotic cell death through the inhibition of the β-catenin and NF-κB signaling pathways in HeLa and ME-180 cells, respectively (Tang et al., 2017; Zhang et al., 2020a). Previous studies have demonstrated that punicalagin can also induce autophagy (Cheng et al., 2016; Wang et al., 2013). The activation of LC3-II conversion, beclin-1 expression, and p62 degradation by punicalagin can trigger autophagy in BCPAP papillary thyroid carcinoma cells (Cheng et al., 2016). Punicalagin use resulted in the induction of apoptosis and autophagy in U87MG glioma cells (Wang et al., 2013). However, the mechanism of punicalagin for promoting apoptotic and autophagic cell death in leukemic cells is not known.

We investigated punicalagin’s mechanism for inducing apoptosis and autophagy in NB4 and MOLT-4 leukemic cells. We demonstrated that punicalagin inhibited cell proliferation and induced cell death through apoptotic and autophagic mechanisms. Punicalagin treatment resulted in the activation of caspase-3/-8/-9, the alteration of Bax and Bcl-2, and the regulation of mTOR/ULK1 signaling to induce apoptosis and autophagy.

Materials & Methods

Leukemic cell culture

NB4 (acute promyelocytic leukemia cell line) and MOLT-4 (acute lymphocytic leukemia cell line) were purchased from Cell Lines Service (Eppelheim, Germany). Cells were cultured in RPMI-1640 medium supplemented with 1% antibiotics (100 U/ml penicillin and 100 µg/ml streptomycin) and 10% (v/v) fetal bovine serum (Gibco Life Technologies, Walthem, MA, USA) at 37 °C with 5% CO2.

Peripheral blood mononuclear cell (PBMC) isolation

Human blood samples were collected with the ethical approval of the Mahidol University Central Institutional Review Board (MU-CIRB) (approvals no. MU-CIRB 2019/310.2911) and written informed consent was obtained from all study participants. Peripheral blood mononuclear cells (PBMCs) were isolated through density gradient centrifugation using Lymphoprep™ (Alere Technology AS, Oslo, Norway). Blood samples were diluted with phosphate buffer saline (PBS) at a 1:1 ratio and the diluted blood was layered over Lymphoprep then centrifuged for 30 min at 800g. The PBMCs were washed twice with medium after harvesting.

Analysis of cell viability assay by MTS assay

Leukemic cells (1 ×104 cells/well) were treated with various concentrations of punicalagin (25, 50, 75 and 100 µg/ml; Sigma-Aldrich, Schnelldorf, Germany). Punicalagin was dissolved in deionized water. We incubated 20 µl of Cell Titer 96 Cell Proliferation Assay Solution (Promega Corp., Madison, WI, USA) for 24 and 48 h before adding the solution to the wells and incubating the samples at 37 °C for 4 h. The absorbance of the formazan product was measured at 490 nm using a spectrophotometer with GEN5™ analysis software (BioTek Instruments, Inc., Winooski, VT, USA). Cell viability was evaluated using untreated cells as the control and then the half-maximal inhibitory concentration (IC50) was calculated for further experiments.

Leukemic cells (1 ×104 cells/well) and PBMC (1 ×105 cells/well) were treated with IC50 concentration of punicalagin (60 and 55 µg/ml for 24 and 48 h, respectively) and/or IC50 of daunorubicin (0.5 and 0.25 µg/ml for 24 and 48 h) (TOKU-E, Bellingham, WA, USA), respectively, to determine the synergistic property of punicalagin and daunorubicin. Cell viability was determined using MTS assay. The combination index (CI) was used to analyze the synergistic effect of punicalagin and daunorubicin (Chou & Talalay, 1984). Synergy, additive effect and antagonism are indicated by CI < 1, CI = 1 and CI > 1, respectively. The CI is calculated using the following equation: Combination indexCI=D1Dχ1+D2Dχ2

where (D)1 and (D)2 are concentrations of first compound and second compound that achieve X% inhibition in the combination; (Dχ)1 and (Dχ)2 are the concentrations of the first compound (alone) and the second compound (alone), respectively, that produce the same effect.

Analysis of cell apoptosis by Annexin V-FITC/PI staining

Apoptotic cells were quantified using the Annexin V assay kit (BD Biosciences, Palo Alto, CA, USA). The cells (1 ×105 cells/well) were treated with IC50 concentration of punicalagin (55 µg/ml) for 48 h and were then washed twice with PBS. The cells were centrifuged and resuspended in a binding buffer before being stained with Annexin V/FITC and Propidium iodide (PI) for 15 min in the dark. Data from 10,000 events per sample were collected and apoptotic cells were measured using a FACScantoII flow cytometer (BD Biosciences).

Analysis of induction of autophagy by anti-LC3/FITC antibodies staining

The Guava® Autophagy LC3 Antibody-based Detection Kit (Luminex Corporation, Austin, TX, USA) was used to determine punicalagin-induced autophagy. The cells were treated with 55 µg/ml of punicalagin for 48 h before being incubated with diluted Autophagy Reagent A, which is a lysosomal inhibitor. The cells were centrifuged and resuspended in Autophagy Reagent B. The cells were stained with FITC-conjugated anti-LC3 antibodies. LC3-II mean fluorescence intensity can be used to quantify autophagosomes and this was measured using flow cytometry.

Determination of gene expression by reverse transcription- quantitative PCR (RT-qPCR)

Leukemic cells were treated with punicalagin at IC50 concentration for 48 h. Total RNA was extracted by Genezol™ reagent (New England Biolab, Inc., Ipswich, MA, USA) according to the manufacturer’s recommendations. First strand cDNA was synthesized by the reverse transcription of 2 µg template RNA using the RevertAid first strand cDNA synthesis kit (Themo Scientific, Waltham, MA, USA). cDNA was then amplified by Luna real time PCR master mix (New England Biolan Inc., Ipswich, MA, USA). The primers were designed from literature reviews and their quality was determined by the BLASTN database (Chatupheeraphat et al., 2020; Deesrisak et al., 2021). The primer sequences are shown in Table 1. Real-time PCR was performed with the Bio-Rad thermal cycle CX1000 (Bio-Rad, Inc., Hercules, CA, USA). mRNA expression was normalized to internal control GAPDH. The gene expression was calculated using the 2−ΔΔct method (Livak & Schmittgen, 2001).

Table 1 The sequence of primers used in this study.

Primers	Sequence	
Caspase-3	F: 5′-TTCAGAGGGGATCGTTGTAGAAGTC-3′
R: 5′-CAAGCTTGTCGGCATACTGTTTCAG-3′	
Caspase-8	F: 5′-GATCAAGCCCCACGATGAC-3′
R: 5′-CCTGTCCATCAGTGCCATAG-3′	
Caspase-9	F: 5′-CATTTCATGGTGGAGGTGAAG-3′
R: 5′-GGGAACTGCAGGTGGCTG-3′	
Bax	F: 5′-CGAGAGGTCTTTTTCCGAGTG-3′
R: 5′-GTGGGCGTCCCAAAGTAGG-3′	
Bcl-2	F: 5′-ATGTGTGTGGAGAGCGTCAA-3′
R: 5′-GCCGTACAGTTCCACAAAGG-3′	
mTOR	F: 5′-CGCTGTCATCCCTTTATCG-3′
R: 5′-ATGCTCAAACACCTCCACC-3′	
ULK1	F: 5-′GGCAAGTTCGAGTTCTCCCG-3′
R: 5′-CGACCTCCAAATCGTGCTTCT-3′	
GAPDH	F: 5′-GCACCGTCAAGGCTGAGAA-3′
R: 5′-AGGTCCACCACTGACACGTTG-3′	

Construction of protein-chemical interaction networks by bioinformatic analysis

The interaction network between punicalagin and its targeted proteins was predicted using the STITCH database. The inputs for analysis were composed of punicalagin, apoptotic proteins (CASP3, CASP8, CASP9, BAX and BCL2) and autophagic proteins (MTOR and ULK1) for Homo sapiens with a medium confidence (0.400).

Statistical analysis

The experiments were performed in triplicate. Data were expressed as mean ± standard error of mean (S.E.M). The results between the two groups were analyzed using the student’s t-test and comparisons of more than two groups were performed by one-way ANOVA using SPSS (SPSS Inc., Chicago, IL, USA). A statistically significant difference was considered as p-value < 0.05.

Results

Effect of punicalagin on the proliferation of leukemic cells

Leukemic cells (NB4 and MOLT-4) were treated with various concentrations of punicalagin (25, 50, 75 and 100 µg/ml) for 24 and 48 h to investigate the effects of punicalagin on cell viability. Their viability was examined using the MTS assay. Punicalagin significantly decreased the viability of NB4 and MOLT-4 cells in a dose-dependent manner (Figs. 2A and 2B). The comparative results between punicalagin-treated cells and control cells are presented in Table 2. The IC50 of punicalagin was 57.1 and 53.5 µg/ml in NB4 cells after 24 and 48 h of treatment as well as 65.7 and 58.9 µg/ml in MOLT-4 cells after 24 and 48 h of treatment, respectively. The IC50 value at 48 h was selected for further experiment as it showed higher cytotoxic effect compared with 24 h. The results suggested that punicalagin inhibited NB4 and MOLT-4 leukemic cells.

Figure 2 Punicalagin inhibits the cell viability of leukemic cells.

(A) NB4 and (B) MOLT-4 leukemic cells were treated with various concentrations of punicalagin for 24 and 48 h. Cell viability was measured using MTS assay. *p < 0.05 was considered to be a statistically significant difference from the control.

Table 2 Comparative statistics of viability between punicalagin-treated cells and control cells.

Group	%Cell viability of NB4
(mean ± S.E.M)	p-value	Group	%Cell viability of MOLT-4
(mean ± S.E.M)	p-value	
24 h	
Control	100.00 ± 0.00	–	Control	100.00 ± 0.00	–	
Punicalagin
(µg/ml)	25	77.50 ± 1.26	0.2190	Punicalagin
(µg/ml)	25	86.42 ± 3.22	0.3743	
50	54.13 ± 2.39	0.0058	50	73.17 ± 8.22	0.0238	
75	31.02 ± 8.81	0.0003	75	40.83 ± 6.99	0.00007	
100	17.37 ± 2.21	0.00005	100	16.53 ± 0.56	0.000003	
48 h	
Control	100.00 ± 0.00	–	Control	100.00 ± 0.00	–	
Punicalagin
(µg/ml)	25	79.06 ± 1.26	0.1234	Punicalagin
(µg/ml)	25	78.66 ± 1.92	0.0175	
50	48.07 ± 8.85	0.0004	50	61.56 ± 5.81	0.0002	
75	30.62 ± 7.91	0.00003	75	34.29 ± 5.65	0.0000019	
100	8.65 ± 2.78	0.000003	100	14.17 ± 1.68	0.0000002	
Notes.

Significantly different for p-values < 0.05 indicated in bold.

Effect of the combination of punicalagin and daunorubicin on leukemic cells

Cell viability was assessed using the MTS assay to determine the effect of punicalagin combined with a chemotherapeutic drug. NB4 and MOLT-4 cells were first treated with 0.25, 0.5, 0.75 and 1 µg/ml of the chemotherapeutic drug, daunorubicin, before being incubated for 24 and 48 h. Daunorubicin decreased the viability of NB4 and MOLT-4 cells in a dose-dependent manner (Figs. 3A and 3B). The comparisons of cell viability between daunorubicin-treated leukemic cells and control cells are presented in Table 3. NB4 and MOLT-4 were then treated with IC50 of punicalagin, daunorubicin alone, or the combination of both compounds for 24 and 48 h. The combination of punicalagin and daunorubicin significantly decreased cell viability compared with punicalagin or daunorubicin treatment alone (Fig. 4). The effect on PBMCs was also assessed. This combination significantly decreased cell viability of PBMCs but had less effect than NB4 and MOLT-4 leukemic cells. The comparative statistical analyses (p-value) of the combination results were presented in Table 4 and the CI was used to determine the synergistic effect of punicalagin and daunorubicin. The results showed that the CI of NB4 cells after 48 h of treatment, and the MOLT-4 cells after 24 and 48 h of treatments, were 0.80, 0.94 and 0.81, respectively, which represent a synergistic effect. However, the CI of NB4 cells after 24 h of treatment was 1.02, which indicates an additive effect. These results suggested that punicalagin enhanced the cytotoxicity of daunorubicin in NB4 and MOLT-4.

Figure 3 Effect of daunorubicin on cell viability in leukemic cell lines.

(A) NB4 and (B) MOLT-4 leukemic cells were treated with various concentrations of daunorubicin for 24 and 48 h. Cell viability was determined by MTS assay. *p < 0.05 was considered to be a statistically significant difference from the control.

Table 3 Statistical analysis of cell viability between daunorubicin-treated leukemic cells and control cells.

Group	%Cell viability of NB4
(mean ± S.E.M)	p-value	Group	%Cell viability of MOLT-4
(mean ± S.E.M)	p-value	
24 h	
Control	100.00 ± 0.00	-	Control	100.00 ± 0.00	-	
Daunorubicin (µg/ml)	0.25	77.31 ± 5.55	0.0095	Daunorubicin (µg/ml)	0.25	84.85 ± 1.35	0.021	
0.5	47.99 ± 5.58	0.0000121	0.5	53.14 ± 1.61	0.0000025	
0.75	33.27 ± 1.21	0.0000012	0.75	40.79 ± 4.74	0.0000003	
1.0	30.60 ± 0.93	0.0000008	1.0	32.40 ± 3.45	0.0000001	
48 h	
Control	100.00 ± 0.00	-	Control	100.00 ± 0.00	-	
Daunorubicin (µg/ml)	0.25	52.61 ± 6.92	0.0001	Daunorubicin (µg/ml)	0.25	61.04 ± 3.57	0.0000876	
0.5	22.35 ± 6.57	0.0000012	0.5	22.37 ± 6.53	0.0000001	
0.75	15.94 ± 0.44	0.0000006	0.75	15.99 ± 1.53	0.0000001	
1.0	14.75 ± 0.06	0.0000005	1.0	14.27 ± 1.13	0.0000001	
Notes.

Significantly different for p-values < 0.05 indicated in bold.

Figure 4 The synergistic effect of punicalagin and daunorubicin.

NB4, MOLT-4, and peripheral blood mononuclear cells (PBMC) were treated with IC50 of punicalagin with or without daunorubicin for (A) 24 h and (B) 48 h. Cell viability was measured using MTS assay. *p < 0.05 was considered to be a statistically significant difference significant different from the punicalagin or daunorubicin treatment alone.

Punicalagin induced apoptosis in leukemic cells

To determine the effect of punicalagin on leukemic cell apoptosis, NB4 and MOLT-4 cells were treated with IC50 concentration of punicalagin. The percentage of apoptotic cells was measured using Annexin V-FITC/PI staining after 48 h of incubation. Punicalagin significantly increased the total percentage of apoptotic cells in NB4 and MOLT-4 compared with untreated cells as shown in Figs. 5A–5C (p = 0.0003 and 0.0000001, respectively). The results show punicalagin induced apoptosis in NB4 and MOLT-4 cells.

Punicalagin increased the LC3-II level for autophagy induction

NB4 and MOLT-4 leukemic cells were treated with IC50 of punicalagin for 48 h to evaluate the effect of punicalagin on autophagy. Intracellular autophagosome staining was used to detect autophagic activity. The LC3-II mean fluorescence intensity (MFI), which quantifies autophagosomes, was analyzed by anti-LC3 conjugated FITC antibody staining. The results showed that punicalagin significantly increased LC3-II MFI in NB4 and MOLT-4 compared with the control (Figs. 6A–6C) (p = 0.00002 and 0.00002, respectively). The results suggest that punicalagin induced autophagy in NB4 and MOLT-4 cells.

Table 4 Comparative statistics of the combination result.

Cell	Group	24 h	48 h	
		% Cell viability
(mean ± S.E.M)	p-value	% Cell viability
(mean ± S.E.M)	p-value	
NB4	The combinationa	29.66 ± 2.605	0.028	20.26 ± 0.358	0.044	
Punicalagin (IC50)	56.01 ± 7.332	52.78 ± 7.044	
The combinationa	29.66 ± 2.605	0.044	20.26 ± 0.358	0.011	
Daunorubicin (IC50)	48.33 ± 5.885	52.69 ± 3.458	
MOLT-4	The combinationa	30.67 ± 3.995	0.039	25.51 ± 2.786	0.008	
Punicalagin (IC50)	54.28 ± 6.744	58.59 ± 6.038	
The combinationa	30.67 ± 3.995	0.009	25.51 ± 2.786	0.002	
Daunorubicin (IC50)	53.18 ± 2.488	54.30 ± 2.607	
PBMC	The combinationa	63.43 ± 2.149	0.002	67.53 ± 4.966	0.016	
Punicalagin (IC50)	85.24 ± 2.306	90.25 ± 2.742	
The combinationa	63.43 ± 2.149	0.032	67.53 ± 4.966	0.045	
Daunorubicin (IC50)	83.59 ± 5.868	88.74 ± 5.476	
PBMC	The combinationa	63.43 ± 2.149	0.001	67.53 ± 4.966	0.001	
NB4	29.66 ± 2.605	20.26 ± 0.358	
PBMC	The combinationa	63.43 ± 2.149	0.002	67.53 ± 4.966	0.002	
MOLT-4	30.67 ± 3.995	25.51 ± 2.786	
Notes.

a The combination containing IC50 of punicalagin and IC50 of daunorubicin.

b Significantly different for p-values < 0.05 indicated in bold.

Figure 5 Effect of punicalagin on early and late apoptotic cells.

NB4 and MOLT-4 cells were treated with IC50 of punicalagin for 48 h and the percentage of apoptotic cells was analyzed using flow cytometry. Scatter plots of Annexin V-FITC/PI stained (A) NB4 and (B) MOLT-4 cells were treated with or without punicalagin. (C) Quantitative results of early and late apoptotic cells. Data were expressed as the mean ± SEM of three independent experiments. *p < 0.05 was considered to be a statistically significant difference from the control group.

Figure 6 Punicalagin promotes autophagy by increasing LC3-II level.

(A) NB4 and (B) MOLT-4 cells were treated with IC50 of punicalagin for 48 h and analyzed LC3-II MFI using flow cytometry. (C) Quantitative result of LC3-II level. *p < 0.05 was considered to be a statistically significant difference from the control group.

Punicalagin regulated apoptotic and autophagic gene expressions in NB4 and MOLT-4 leukemic cells

We examined the alteration of apoptotic mRNA expression, caspase-3, caspase-8, caspase-9, Bax and Bcl-2, and autophagic mRNA expression of mTOR and ULK1 to investigate the mechanism involved in apoptosis and autophagy induction by punicalagin. NB4 and MOLT-4 cells were treated with IC50 of punicalagin for 48 h, which resulted in the mRNA expression of caspase-3, caspase-8 and caspase-9 to be significantly upregulated (p = 0.023, 0.00001 and 0.019 for NB4; p = 0.022, 0.026 and 0.01 for MOLT-4, respectively). These results indicate caspase activation as shown in Fig. 7A. The B-cell lymphoma 2 (Bcl-2) family is an apoptosis regulator that consists of both pro-apoptotic genes, such as Bax, and anti-apoptotic genes, such as Bcl-2. The results showed that punicalagin upregulated the expression of Bax (p = 0.001 for NB4; p = 0.00009 for MOLT-4) and downregulated the expression of Bcl-2 (p = 0.023 and 0.036 for NB4 and MOLT-4, respectively) as shown in Fig. 7B. Moreover, mTOR mRNA expression was significantly decreased (p = 0.002 and 0.002 for NB4 and MOLT-4 cells, respectively) whereas ULK1 mRNA expression was significantly increased (p = 0.036 for NB4; p = 0.045 for MOLT-4) in punicalagin-treated cells (Fig. 7C).

Figure 7 Effect of punicalagin on mRNA expression of apoptotic and autophagic genes.

Leukemic cells were treated with IC50 of punicalagin for 48 h. The mRNA expression of (A) caspase family genes, (B) Bcl-2 family as well as (C) ULK1 and mTOR were determined using reverse transcription-quantitative PCR. *p < 0.05 was considered to be a statistically significant difference from the control group.

Prediction of protein-chemical interactions

Protein-chemical interactions were obtained from STITCH databases and the results showed the interaction between punicalagin and the related proteins. Analysis revealed that punicalagin was related to proteins which play a role in induction of apoptosis (CASP3, CASP8, CASP9, BAX, BCL2, BCL2L11 and APAF1) and mTOR signaling in autophagy regulation (MTOR, ULK1, RPTOR, RICTOR, RPS6KB1, MLST8, EIF4EBP1, FKBP1A and MAPKAP1) via cyclooxygenase-2 (PTGS2) and toll-like receptor 4 (TLR4) (Fig. 8). We determined that caspase-3/-8/-9, Bax, Bcl-2, and mTOR and ULK1 play roles in apoptotic and autophagic signaling pathways in punicalagin treatment.

Figure 8 Construction of protein-chemical interaction using STITCH.

Interactions between protein and chemicals are represented by green lines while protein-protein interactions are represented by grey lines. Stronger associations are shown by thicker lines. CASP3, caspase- 3; CASP8, caspase-8; CASP9, caspase-9; BAX, BCL2-associated X protein; BCL2, B-cell CLL/lymphoma; MTOR, mechanistic target of rapamycin; ULK1, unc-51-like kinase 1; PTGS2, prostaglandin-endoperoxide synthase 2 or cyclooxygenase 2; TLR4, toll-like receptor 4; RPTOR, regulatory associated protein of MTOR; RICTOR, RPTOR independent companion of MTOR; RPS6KB1, ribosomal protein S6 kinase; MLST8, MTOR associated protein; AKT1, v-akt murine thymoma viral oncogene homolog; EIF4EBP1, eukaryotic translation initiation factor 4E binding protein 1; FKBP1A, FK506 binding protein 1A; MAPKAP1, mitogen-activated protein kinase associated protein 1; APAF1, apoptotic protease activating factor 1; BCL2L11, BCL2-like 11.

Discussion

Punicalagin is a major polyphenolic compound found in pomegranate. It reportedly has several biological properties, including antioxidant (Abid et al., 2017), anti-inflammatory (BenSaad et al., 2017) and anticancer properties (Adaramoye et al., 2017; Berköz & Krośniak, 2020; Larrosa, Tomás-Barberán & Espín, 2006; Tang et al., 2016). We studied the anti-leukemic effects of punicalagin and the mechanism underlying the induction of apoptosis and autophagy on NB4 and MOLT-4 leukemic cells. Punicalagin suppressed the proliferation of NB4 and MOLT-4 cells. The cytotoxic effect of punicalagin exhibited a nonsignificant increase on NB4 compared with MOLT-4 cells. Punicalagin was also found to exhibit a synergistic effect with daunorubicin in this study. The combination improved the cytotoxic effect on leukemic cells with low cytotoxicity on healthy PBMC. Punicalagin was also reported to exert a low antiproliferative effect on PBMC compared with HL-60 leukemic cells (Chen et al., 2009). Natural products in combination with chemotherapy drugs have been shown to reduce adverse effects and improve the therapeutic effects of chemotherapy drugs in cancer treatment (Zhang et al., 2020b). Punicalagin was shown to enhance the cytotoxicity of daunorubicin, suggesting that punicalagin may be used in combination with conventional chemotherapy drugs. The mechanism of actions of punicalagin and daunorubicin are still unclear. Daunorubicin is able to intercalate DNA strands to inhibit DNA and RNA synthesis and the topoisomerase II enzyme, and produce reactive oxygen species (ROS) leading to DNA damage and apoptosis (Al-Aamri et al., 2019). It is possible that punicalagin’s mechanism in apoptosis induction may function via the caspase pathway and autophagy via the mTOR/ULK1 pathway. The results of the combined treatment may be related to the similar caspase pathway. Further studies on the side effects of combining punicalagin and daunorubicin treatment are required for determining its role in the inhibition of cell viability.

Apoptosis or type I programmed cell death is characterized by caspase-dependent proteolytic activation, membrane blebbing, and DNA fragmentation. The activation of apoptosis eliminates cancer cells and it is the therapeutic approach for the development of anti-cancer drugs (Carneiro & El-Deiry, 2020). The activation of caspase-8 and caspase-9 plays critical roles in the extrinsic and intrinsic apoptotic pathway, respectively. The active caspase-8 or caspase-9 further activate caspase-3, leading to apoptosis (Mishra et al., 2018). Our results revealed that punicalagin increased the mRNA expression of caspase-3, caspase-8 and caspase-9. Moreover, the Bcl-2 family proteins are mitochondrial apoptotic regulators which can be either pro-apoptotic proteins, such as Bax, or anti-apoptotic proteins, such as Bcl-2 (Mishra et al., 2018). Punicalagin was shown to upregulate Bax mRNA expression and downregulate Bcl-2 mRNA expression in this study. Therefore, our results indicated that punicalagin induced apoptosis via intrinsic and extrinsic apoptotic pathways by activating caspase-3/-8-9 and altering Bax/Bcl-2 in NB4 and MOLT-4 cells. Punicalagin-induced apoptosis has been demonstrated in Caco-2 colon, PC-3 prostate and HeLa cervical cancer cells (Adaramoye et al., 2017; Larrosa, Tomás-Barberán & Espín, 2006; Zhang et al., 2020a). The induction of apoptosis by natural products has been reported in leukemic cell lines (Bozok Cetintas et al., 2014; Debnath et al., 2019). Capsaicin is a component of red chili peppers that has been shown to upregulate the expression of caspase gene family members, activate caspase-3 activity and downregulate the expression of Bcl-2 to induce apoptosis in CCRF-CEM cells (Bozok Cetintas et al., 2014). Bromelain, a pineapple enzyme, in combination with peroxidase, suppressed K562 cell proliferation and promoted the intrinsic pathway of apoptosis through the alteration of the mitochondrial membrane and the regulation of the expression of apoptosis-related proteins, including Bax, Bcl-2, caspase-3 and cytochrome c (Debnath et al., 2019).

Autophagy plays an important role in tumor suppression through the elimination of damaged organelles or proteins, the inhibition of the survival of cancer cells, and the induction of cell death. Several proteins, including AMP-activated protein kinase (AMPK), mammalian target of rapamycin (mTOR), Unc-51-like autophagy-activating kinase (ULK), autophagy-related protein (ATG) and beclin-1, are involved in the regulation of autophagy (Yun & Lee, 2018). Previous studies demonstrated that punicalagin exhibits anticancer activities via the induction of autophagy, which plays a role in cancer cell death (Cheng et al., 2016; Wang et al., 2013). The upregulation of mTOR inhibits autophagy resulting in the stimulation of cancer growth and progression. Thus, mTOR inhibitors have been developed for cancer therapy (Hua et al., 2019). The inhibition of mTOR promotes ULK1 activation which is required for autophagy induction (Kim et al., 2011). Resveratrol, a polyphenolic compound found in berries and grapes, induces autophagy by inhibiting mTOR and activating ULK1 activity (Park et al., 2016). Results from the present study have shown that punicalagin induced autophagy through the increased expression of LC3-II as well as the downregulation of mTOR expression and the upregulation of ULK1 expression. Alterations in gene expression may correlate with protein levels. However, gene expression does not prove the existence of the proteins. The effects of punicalagin treatment on protein expression have been investigated (Ganesan et al., 2020; Wang et al., 2013). Immunoblot analysis demonstrated that the expression of the Bcl-2 protein was decreased while the expression of cleaved caspase-9 and cleaved poly (ADP-ribose) polymerase (PARP) were increased by punicalagin treatment. The increased expression of LC3-II and the phosphorylation of AMPK and p27 were associated with autophagy induction (Wang et al., 2013). Furthermore, the expression of 35 different apoptosis/autophagy-related proteins have been studied using proteome profiling analysis. The results revealed that the altered protein expression of heat shock protein (HSP) 27, HSP 60, catalase, tumor necrosis factor receptor 1/tumor necrosis factor receptor superfamily member 1A (TNFRI/ TNFRSF1A), p53, Bax, Bcl-2, Smac/Diablo, HSP 70, p21 and p27 play roles in the activation or inhibition of apoptosis and/or autophagy in punicalagin-treated HCT 116 cells (Ganesan et al., 2020).

The interaction between punicalagin and the target proteins was constructed using the STITCH bioinformatics tool for an increased understanding the molecular mechanism of apoptosis and autophagy in response to punicalagin. PTGS2 (Cyclooxygenase-2; COX-2) and toll-like receptor 4 (TLR4) were involved in the punicalagin-related molecules interaction network. COX-2, a proinflammatory enzyme required for prostaglandins synthesis, is overexpressed in inflammatory diseases and cancer. COX-2 increases mutated cell proliferation and modulates programmed cell death to promote cancer progression in several cancer models (Liu, Qu & Yan, 2015; Sobolewski et al., 2010). COX-2 is able to prevent apoptosis by increasing the expression of surviving Mcl-2 and Bcl-2 anti-apoptotic proteins by decreasing caspase-3 expression (Goradel et al., 2019). The suppression of COX-2 is associated with autophagy leading to neuronal cell death (Niranjan, Mishra & Thakur, 2018). Several studies have reported natural compounds for cancer chemoprevention that have the potential to suppress COX-2 expression (Cerella et al., 2010). For example, curcumin, a flavonoid from Curcuma longa, inhibits COX-2 and its downstream genes resulting in the induction of apoptosis through the regulation of AMP-activated protein kinase (AMPK) in MCF-7 breast cancer cells and HT-29 colon cancer cells (Mortezaee et al., 2019). Moreover, COX-2 protein expression is inhibited by punicalagin in HT-29 colon cancer cells and mice brain (Adams et al., 2006; Kim et al., 2017). Toll-like receptor 4 (TLR4) transmembrane receptors play key roles in the inflammatory immune response (Hao et al., 2018) and were present in the punicalagin-related proteins network. TLR4 expression reportedly increased in various tumor cells including those involved in the development of non-small cell lung cancer (Wang et al., 2017), breast cancer (Yang et al., 2014) and hepatocarcinoma (Gong et al., 2013). STITCH is a useful resource that incorporates data from metabolic pathways, crystal structures, binding experiments and drug–target relationships (Kuhn et al., 2008). It provides generalized analysis for the overview of the interaction between the chemical and its targets and is not a cell type-specific tool. Our results indicate that the protein interactions between punicalagin and its targets in NB4 and MOLT-4 cells should be considered in future research.

Conclusions

Our study revealed that punicalagin exerted anti-leukemic effects by suppressing cell proliferation and promoting apoptotic and autophagic cell death by regulating caspase, Bax/Bcl-2 and the mTOR/ULK1 signaling pathway. The findings of this study may assist in the creation of complementary and alternative treatments in the future.

Supplemental Information

Supplemental Information 1 Raw data of MTS assay, flow cytometry and RT-qPCR

Click here for additional data file.

Additional Information and Declarations

Competing Interests

Author Contributions

Human Ethics

Data Availability

The authors declare there are no competing interests.

Paweena Subkorn conceived and designed the experiments, performed the experiments, analyzed the data, prepared figures and/or tables, authored or reviewed drafts of the paper, and approved the final draft.

Chosita Norkaew performed the experiments, prepared figures and/or tables, and approved the final draft.

Kamolchanok Deesrisak analyzed the data, prepared figures and/or tables, and approved the final draft.

Dalina Tanyong conceived and designed the experiments, authored or reviewed drafts of the paper, and approved the final draft.

The following information was supplied relating to ethical approvals (i.e., approving body and any reference numbers):

Mahidol University Central Institutional Review Board (MU-CIRB) approved this research (approval no. MU-CIRB 2019/310.2911).

The following information was supplied regarding data availability:

The raw measurements are available in the Supplemental Files.

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
