# Peer review of "Punicalagin, a pomegranate compound, induces apoptosis and autophagy in acute leukemia"

_PeerJ, doi:10.7717/peerj.12303_

## Round 0.1 · original submission · Major Revisions

Please address the issues highlighted by our reviewers. I remind you that, although novelty is not necessarily a requirement in PeerJ, we do require every paper to be a significant addition to the literature. Therefore, you should also sufficiently address how your report adds to the information presented earlier in Dahlawi, H. 2013 J SCI NUTR 1(2): 196-208.

·

Basic reporting

No comment

Experimental design

No comment

Validity of the findings

No comment

Additional comments

This paper reports data on antileukaemic effects of punicalagin, being a pomegranate compound, evaluated in in vitro studies with the use of NB4 and MOLT-4 leukemic cell lines. Although the manuscript presents some potentially interesting results, there are a number of points concerning their presentation and discussion that need to be addressed.

Specific points
1. “Introduction” and Discussion” sections require some reediting to avoid repetitive information found in both sections.
2. It would be worth to include an additional figure presenting the chemical structure of punicalagin.
3. “Materials and methods” section – providers of punicalagin, daunorubicin and some chemicals used in the study should be given.
4. “Results” section, lines 157-162, 163-164 and “Discussion” section, lines 231-232 – to demonstrate that punicalagin acts synergistically with daunorubicin, an isobolographic analysis should be performed.
5. “Results” section, line 163 – the statement: “The normal PBMCs were less affected than leukemic cells” requires more detailed comment.
6. “Discussion” section, lines 213-214 – the p-values should be shown to demonstrate the correctness of the statement that the cytotoxic effect of punicalagin was higher on NB4 than MOLT-4 cells.
7. Some sentences/terms require careful revision, e.g. i. Abstract, lines 22-23: “The mRNA expression of apoptosis and autophagy was determined using reverse transcriptase-quantitative PCR”; ii. “Materials and methods” section, line 110 : “… apoptosis cells were measured by FACScantoII flow cytometer …”; iii. “Results” section, line 183: “… we examined the alteration of apoptotic genes”; iv. legend to Figure 6: “ … mRNA expression level …”.
8. Figure 4C – given the observed differences in apoptotic effects of punicalagin on NB4 and MOLT-4 cells, it would be more interesting to present the percentages of early apoptotic, late apoptotic and necrotic cells, respectively rather than just showing the total percentage of apoptotic cells.
9. Some English and editorial work should be performed.

Minor points
1. Table 2 and Table 3 – it should be added that the “mean ± S.E.M” values are given in %.
2. Legend to Figure 7 – the given full name of APAF-1: “apoptotic peptidase activating factor 1” should be replaced by “apoptotic protease activating factor 1”.

Reviewer 2 ·

Basic reporting

No comment.

Experimental design

1) The in vitro anti cancer properties of phenolic compounds contained in pomegranate juice and in particular of punicalagin have benn largely related in the past. In particular, the cytotoxic and proapoptotic effect of punicalagin on MOLT-4 cell line, one of the two leukemic models described here, have already been reported (Dahlawi, H. 2013 J SCI NUTR 1(2): 196-208).
2) The authors should provide some insights into the physiologic relevance of the punicalagin concentrations used. The complex metabolism of ellagitannins in the gastrointestinal tract strongly decreases their bioavailability, rendering their presence in the blood very unlikely. (Espin, JC 2007 J Agric Food Chem 55(25):10476-10485).
3) In the methods section, the authors do not mention in which buffer the chemical compounds were dissolved.
4) Related to the previous point, it is not mentioned whether the in the control groups the cells were treated with vector buffer at similar concentration as in the treatments groups, or whether cells were just cultured in normal culture medium.
5) The authors do not provide any information about the design of the primers used (home-designed or from literature, in this case cite sources), on their amplifications efficiencies (an essential data to assess whether the deltadelta Ct method can be applied without efficiency corrections) and on stability of the housekeeping gene across treatments.
6) The STITCH analysis based on such a little set of data may lead to overinterpretations.
7) The term synergy to describe the effect of punicalagin in combination with daunorubicin is misused. Indeed, the cytotoxic effect of the two compounds used in combination is evel lower than the sum of the two used separately.
8) in the first figure the authors should discuss the absence of a significant difference between 24 and 48 hours of tretments at similar concentrations.

Validity of the findings

The data presented clearly demonstrate that treatment of cells with punicalagin, alone or in combînation with chemotherapeutic drug, induces NB4 and MOLT4 cell lines apoptosis and autophagy. Nevertheless, the insights provided in this study, as compared to the state of the art, is very limited. The authors omitted to discuss the relevance of the experimental conditions applied and the suggestion to use punicalagin as an anticancer drug, in this case for the treatment of leukemia, based on the presented data, exceeds the level of simple and reasonable speculation and should be omitted.

Additional comments

The topic covered by the study presented in this article is interesting, but researchers active in the field of polyphenols effect on different tissues/cells type in vitro should always carefully design their studies considering the bioavailability of the compounds in vivo, in order to remain in a range of physological relevance.

---

## Round 0.2 · Major Revisions

Our original reviewers gave conflicting advice, and therefore I have had to procure a third opinion. Please address all issues pointed out by the adjudicating reviewer (reviewer #3) , as well as the ones earlier mentioned by reviewer #2 and not yet addressed.

·

Basic reporting

No comment

Experimental design

No comment

Validity of the findings

No comment

Additional comments

In the revised version of the paper the Authors included modifications that followed all detailed remarks and suggestions of the reviewer. The revised version of the manuscript may be accepted for publication after suitable linguistic and editorial corrections.

Reviewer 2 ·

Basic reporting

No comment

Experimental design

Compared to the first version, the authors completed the article with some of the information requested. Nevertheless, some of the concerns were not answered. For example, nowhere is mentioned how the primers were designed and no information is provided on the efficiency of the different amplification systems. Moreover, the term synergy is still misused.

Validity of the findings

The work presented does not provide insights into the topic addressed. Changes implemented in the article are not sufficient to render it suitable for publication. Despite previous comments suggesting to eliminate speculations on the possible use of punicalagin in treatment of acute leukemia, the authors insist. In this version, they go even further in speculation suggesting the possible use of deliver vehicle molecules to provide punicalagin to target organs.

Additional comments

The changes introduced in this new version does not provide any qualitative improvement to the article. Both the experimental design and the arguments in the "discussions" section of the results remain questionable.

Reviewer 3 ·

Basic reporting

Sufficient literature review was done, however the findings of different studies were just incorporated in the writing without analyzing and synthesizing them. There is lack of coherence in the flow of the article. Clearer problem statement, research gap and novelty of this study should have been included in the introduction.

Experimental design

No comment

Validity of the findings

The data are presented clearly and in organized manner by the authors. However, the claims of the findings could have been stronger if there was a proteomics study done on the proteins of interest. Given that the main claim of the study is that punicalagin induces apoptosis and autophagy, it is crucial to provide sufficient evidence through relevant experimental design. Some of the important points to consider are given below:

Important Points
1. Combined cytotoxic effect of punicalagin and daunorubicin was studied in MTS but not in the rest of the experiments. Authors must justify the reason to utilize punicalagin (only) treatment for most part of the study instead of combination therapy as tested in MTS.
2. IC50 was achieved for both 24 and 48-hour time points with small difference (eg. 57.1 and 53.5 μg/ml in NB4 cells for 24 and 48 h of treatment respectively). Provided there’s no significant difference in IC50 between the two time points, the authors did not mention why the following experiments only used IC50 of punicalagin achieved at 48 hours.
3. Gene expression does not prove the existence of the proteins (we can’t be so sure if they are translated unless tested) or their concentrations. It is only an indirect conclusion. Hence, to confirm and quantify proteins of interest, western blot or proteome profiling must have been performed. Please refer studies with similar findings that had done proteomic works (Ganesan, Sinniah, Chik, & Alshawsh, 2020; Wang et al., 2013).
4. Analysis through STITCH is a great aiding tool but it cannot be used as an evidence of protein interaction between punicalagin and NB4 or MOLT-4 in this study as STITCH provides a generalized analysis, not cell-specific.
5. The claim that punicalagin enhances effect of daunorubicin (line 234-235) needs to take into consideration of the mechanism of actions of these 2 different compounds and possible combined side effects given that the combination treatment is only studied in MTS in this study and not throughout.
6. Reference for line 251, “The induction of apoptosis by natural products has been reported in leukemic cell” is missing.
7. The discussion part could be elaborated better because there is more literature review than discussion of the results of the study. For example, lines 294-303 are more suitable to be included in the introduction instead of discussion.
8. There are a few grammatical errors and incorrect sentence structures that the authors need to rectify.

Additional comments

The study explores a very interesting subject and the authors have made genuine efforts to achieve the aim of this study. However, more attention should have been given to the study design to be more comprehensive in investigating the apoptotic and autophagic activity of punicalagin in acute leukemia.

---

## Round 0.3 · Minor Revisions

I think your manuscript can be accepted, provided that the language quality is improved: specifically, large portions of the newly incorporated text are quite ungrammatical (such as "need to be concerned.", etc. etc.).

Reviewer 3 ·

Basic reporting

No comments

Experimental design

No comments

Validity of the findings

No comments

Additional comments

No comments

---

## Round 0.4 · accepted · Accept

I am glad to accept your manuscript for publication in PeerJ. Congratulations!